



# Acidification and nutrient management are projected to cause
# reductions in shell and tissue weights of oysters in a coastal plain
# estuary
Catherine R. Czajka[1], Marjorie A.M. Friedrichs[1], Emily B. Rivest[1], Pierre St-Laurent[1], Mark J. Brush[1],
and Fei Da[2]
[1]Virginia Institute of Marine Science, William & Mary, Gloucester Point, VA, 23062, USA
[2]Department of Geosciences, Princeton University, Princeton, NJ, 08544, USA
*Correspondence to*: Catherine R. Czajka (czajkacatherine@gmail.com)
**Abstract.** Coastal acidification, warming, and nutrient management actions all alter water quality conditions that marine
species experience, with potential impacts to their physiological processes. Decreases in calcite saturation state ($\Omega_{Ca}$) and food
availability, combined with warming water temperatures, pose a threat to calcifying organisms; however, the magnitude of
future changes in estuarine systems is challenging to predict and is not well known. This study aims to determine how and
where oysters will be affected by future acidification, warming, and nutrient reductions, and the relative effects of these
stressors. To address these goals, an oyster bioenergetics model for Eastern oysters (*Crassostrea virginica*) was embedded in
a 3-D coupled hydrodynamic-biogeochemistry model implemented for two tributaries in the lower Chesapeake Bay. Model
simulations were forced with projected future conditions (mid-21st century atmospheric $CO_2$, atmospheric temperature, and
managed nutrient reductions) and compared with a realistic present-day reference run. Together, all three stressors are
projected to reduce $\Omega_{Ca}$ and growth of oyster shell and tissue. Increased atmospheric $CO_2$ and temperature are both projected
to cause widespread reductions in $\Omega_{Ca}$. The resulting reductions in oyster shell and tissue growth will be most severe along the
tributary shoals. Future warming during peak oyster growing seasons is projected to have the strongest negative influence on
tissue and shell growth, due to summer water temperatures reducing filtration rates, enhancing shell dissolution and oyster
respiration rates, and increasing organic matter remineralization rates, thus reducing food availability. Nutrient reductions will
exacerbate deficits in oyster food availability, contributing to further reductions in growth. Quantifying the effects of these
stressors provides insight on the areas in the lower bay where oysters will be most vulnerable to mid 21st-century conditions.
**Short summary.** Under future acidification, warming, and nutrient management, substantial reductions in shell and tissue
weights of Eastern oysters are projected for the Chesapeake Bay. Lower oyster growth rates will be largely driven by reduced
calcium carbonate saturation states and reduced food availability. Oyster aquaculture practices in the region will likely be
affected, with site selection becoming increasingly important as impacts will be highly spatially variable.



## 1 Introduction

Anthropogenic climate change and its associated impacts on water quality may threaten marine organisms and
economic systems reliant on them. Oceanic uptake of increasing anthropogenic atmospheric carbon dioxide ($CO_2$) causes a
decrease in seawater pH and saturation states of calcium carbonate, a phenomenon known as ocean acidification (Caldeira and
Wickett, 2003; Doney et al., 2009). Globally, the ocean has absorbed about 30% of anthropogenic atmospheric $CO_2$ since pre-
industrial times (Gruber et al., 2019), and open-ocean surface pH is anticipated to decrease by 0.3 units on average relative to
the 2010s by the end of the century under 'business-as-usual' conditions (Riahi et al., 2011; IPCC, 2019). The percent volume
of ocean water undersaturated with calcite ($\Omega_{Ca} < 1$) is predicted to expand to 91% by 2100 from 76% in the 1990s (Caldeira
and Wicket, 2005; Gattuso et al., 2015).
Since estuaries have lower and more variable pH than the open-ocean, the effects of increased $CO_2$ on estuarine water
quality and biota are often amplified. In coastal and estuarine systems, acidification may be exacerbated by local-level
processes, such as the input of acidic freshwater and nutrient runoff from precipitation, a process termed coastal acidification
(Salisbury et al., 2008; Wallace et al., 2014; Carstensen and Duarte, 2019). Freshwater has relatively low total alkalinity (TA),
or buffering capacity, so areas in estuaries with greater relative freshwater influence cannot resist changes to pH as easily as
more saline or open-ocean waters (Hasler et al., 2018; Pacella et al., 2024). Eutrophication, the increased rate of organic matter
input to a system (Nixon, 1995), may drive large variations in local pH and overall water quality. Elevated nutrient inputs
cause pH to increase in surface waters due to higher primary productivity, which will reduce surface acidification; however,
pH will decrease in deeper bottom waters as the additional organic matter sinks and is remineralized (Cai et al., 2021).
Management actions to reduce eutrophication and improve water quality in bottom waters have been successful but may also
enhance acidification in shallow surface waters by lowering primary productivity (Borges and Gypens, 2010). The overall
effect of future changes in nutrient inputs on coastal biogeochemistry is thus unclear.
Warming, another driver of biogeochemical change in coastal waters, may compound or offset the effects of increased
atmospheric $CO_2$ on coastal ecosystems. The global ocean has absorbed approximately 93% of the atmospheric heat produced
by anthropogenic activity, leading to a global sea surface temperature increase of 0.7°C since 1900 (Jewett and Romanou,
2017). Ocean warming is expected to continue, with global averages increasing by 2.7°C by 2100 and greater increases
expected in shallow coastal regions (Jewett and Romanou, 2017). Coastal acidification may accelerate as warming of coastal
waters increases rates of biogeochemical processes; increased respiration rates may drive larger diel variations in pH, dissolved
oxygen, and associated water quality (Du et al., 2018; Tian et al., 2021). Therefore, it is vital to understand how warming will
interact with acidification to predict local changes in water quality and health of coastal organisms.
Characterizing spatiotemporal patterns of acidification in estuarine waters is important, as negative impacts of
acidification on the biology of marine organisms may be substantial. Acidification disrupts the formation of calcium carbonate
($CaCO_3$) during shell-building, i.e., biocalcification, which is a vital process for growth and survival of many aquatic
invertebrate species (e.g., Orr et al., 2005; Gazeau et al., 2007; Dong et al., 2023). Under acidified conditions, water



concentrations of $CO_2$ and $H^+$ increase, and concentrations of carbonate ions ($[CO_3^{2-}]$) decrease. A low ambient $[CO_3^{2-}]$ inhibits
calcifying organisms from forming $CaCO_3$ for their shells, as more energy is required to precipitate $CO_3^{2-}$ from acidified waters
(e.g., Guinotte and Fabry, 2008; Lutier et al., 2022; Matoo et al., 2020; Mederios and Souza, 2023). Low $\Omega_{Ca}$ may also lead to
net dissolution of $CaCO_3$, leading to weaker shells and greater juvenile susceptibility to predation (e.g., Waldbusser et al.,
2011; Amaral et al., 2012; Barclay et al., 2020). Acidification may further reduce shell growth through adverse physiological
effects that limit energy availability for calcification. Because acidification is often more extreme in estuaries, oysters and
other commercially valuable coastal bivalve species experience stronger effects of climate change than organisms living in
open-ocean environments (Poach et al., 2019; Melzner et al., 2020; Cai et al., 2021; ). Prior experiments have revealed negative
effects of acidification, warming, and nutrient reductions on oyster biocalcification and growth (Beniash et al., 2010;
Waldbusser et al., 2011; Gobler and Talmage, 2014), but it is yet to be determined how the impacts of these stressors on oyster
shell and tissue growth will vary spatially in highly dynamic systems.

The Chesapeake Bay is an excellent study system for examining the interacting influences of acidification, warming,

and nutrient reductions (hereafter referred to collectively as "future stressors") on estuarine biogeochemistry and the organisms
living there. The bay exhibits high temporal and spatial variability in pH due to seasonal phytoplankton blooms, eutrophication,
and acidic freshwater input (Da et al., 2021; St-Laurent et al., 2020; Kemp et al., 2005; Cai et al., 2021). From the mid-1980s
to mid-2010s, surface waters in the upper bay experienced pH increases between +0.2 and +0.4 pH units in early spring and
fall due to increased riverine TA from reduced acid mine drainage and lowered nitrate inputs, while surface waters in the
nitrogen-limited middle bay decreased up to -0.24 pH units during late spring and summer as a result of decreased primary
production (Da et al., 2021). Over the same time period, the bay warmed by $0.24 \pm 0.15°C$ per decade (Hinson et al., 2022),
more than double the average rate of warming for the upper 75m of the global ocean (Rhein et al., 2013). Warming has also
led to more severe hypoxia (Irby et al., 2018; Ni et al., 2020; Frankel et al., 2022; Hinson et al., 2023). In 2010, the
Environmental Protection Agency mandated a Total Maximum Daily Load (TMDL) of pollutants from point and non-point
sources to be achieved by 2025 (EPA, 2010). As nutrient reductions negatively affect pH in surface waters of the bay (Shen et
al., 2020; Da et al., 2021), achieving the TMDLs may actually worsen acidification in shallow and near-shore regions. Much
of the research effort devoted to characterizing present-day carbonate chemistry and its historical trends has focused on the
mainstem and upper Chesapeake Bay (Cai et al., 2017; Shen et al., 2020, Su et al., 2020), and less is known about these
conditions throughout the tributaries of the lower bay (Shadwick et al., 2019).

The combined effects of future stressors will impact calcifying organisms in the lower Chesapeake Bay as well as the

economic systems reliant upon them. The Eastern oyster *Crassostrea virginica* (Gmelin, 1791) is a foundation species native
to the bay (Dayton, 1972). Eastern oyster aquaculture in this region has grown rapidly in the past few decades, with Virginia
becoming the third most productive oyster fishery in 2018 (Hudson, 2019), largely a result of the development of disease-
resistant oyster strains (Frank-Lawale et al., 2014). Negative impacts of acidification on aquaculture practices in other parts of
the world (Barton et al., 2015) have already stirred concern over the vulnerability of oysters in the Chesapeake Bay. For
example, in the Pacific Northwest, major larval mortality occurred at a shellfish hatchery following an upwelling event that



lowered pH and Ω of aragonite, which had cascading impacts on the oyster industry all along the West Coast (Barton et al.,
2015). While most effects of acidification on aquaculture have been observed in oyster larvae in hatcheries, fewer studies have
examined acidification's influence on adult oysters when deployed in the field. Water quality conditions within oyster farms
can be highly spatially variable, so the impacts of acidification may vary with growing conditions (Saavedra et al., 2024;
Simpson et al., 2024). To support the future of the oyster aquaculture industry in Chesapeake Bay, it is critical to identify
which areas in the bay will be most vulnerable to acidification at mid-century and how each driver of change contributes to
acidification and its impacts on growth.
This study addresses the following primary research question: How and where will carbonate chemistry and Eastern
oyster growth in the lower Chesapeake Bay change in the future and which future stressors will drive these changes? A three-
dimensional hydrodynamic-biogeochemical model is coupled with an oyster bioenergetics model and is applied to two major
Virginia tributaries of the Chesapeake Bay. The model provides detailed information on present-day environmental conditions,
and when combined with climate projections from Earth System Models, allows for simulations of the independent and
interacting influences of future environmental change on carbonate chemistry and Eastern oysters. This study provides insight
into which areas are most vulnerable to mid 21st-century acidification and how acidification, warming, and nutrient loading
may each impact oyster growth in isolation as well as via simultaneous co-stressors.
**2 Methods**
**2.1 Model description**
**2.1.1 Hydrodynamic model**
This study uses the three-dimensional hydrodynamic Regional Ocean Modeling System (ROMS; Shchepetkin and
McWilliams, 2005), implemented similarly to St-Laurent and Friedrichs (2024) but on a higher resolution grid focused on two
of the lower Virginia Chesapeake Bay tributaries (Fig. 1). The model domain (Da et al., 2024) includes the York and
Rappahannock Rivers, as well as a portion of the mainstem shoal north of the Rappahannock. The model grid consists of
620x740 horizontal grid cells with a horizontal resolution of 120 m, allowing for greater resolution of coastlines than many
other Chesapeake Bay model grids (Irby et al., 2016). The hydrodynamic model includes 20 terrain-following vertical levels
and two primary state variables: practical salinity and potential temperature. A wetting and drying scheme has been
implemented to represent water levels and currents in coastal grid cells (Warner et al., 2013; St-Laurent and Friedrichs, 2024).






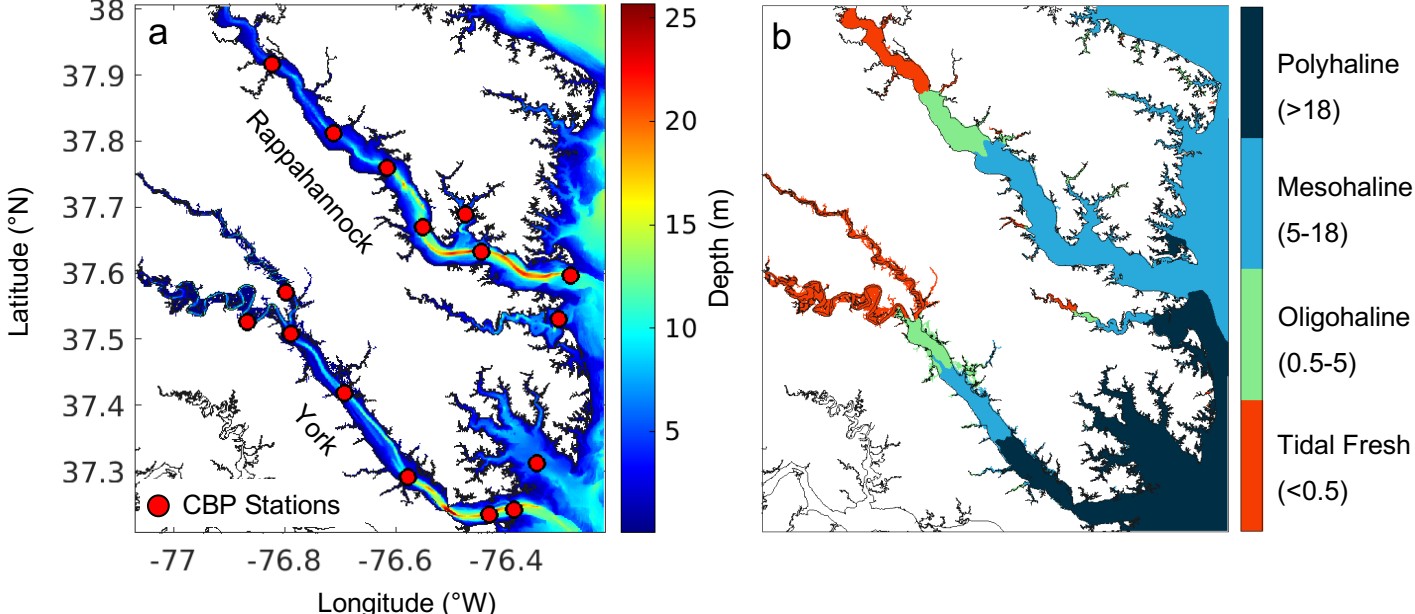

**Figure 1. ROMS-ECBO model domain of Chesapeake Bay tributaries illustrating (a) bathymetry in meters and locations of**
**Chesapeake Bay Program (CBP) water quality monitoring stations (red circles) and (b) bottom salinity zones.**

**2.1.2 Carbon and biogeochemistry model**


The Estuarine-Carbon-Biogeochemistry model (ECB) embedded in ROMS and used in this study has previously been
implemented in the Chesapeake Bay (Feng et al., 2015; St-Laurent et al., 2020; Frankel et al., 2022; Hinson et al., 2023) as
well as in the lower Virginia tributaries (Da et al., 2024). ECB simulates full carbon and nitrogen cycles of the lower trophic
levels, represented by the following state variables: nitrate, ammonium, phytoplankton and zooplankton nitrogen, small and
large detrital nitrogen and carbon, semi-labile and refractory dissolved organic nitrogen, DIC, TA, and dissolved oxygen ($O_2$).
Phytoplankton and zooplankton carbon and dissolved organic carbon (DOC) are calculated from established C:N ratios
(Redfield, 1934; Hopkinson et al., 1998). Biogeochemical processes include primary production, aggregation, sinking, basal
metabolism, exudation, sloppy feeding, excretion, metabolism, nitrification/denitrification, remineralization, grazing, and
mortality. Additional biogeochemical sources and sinks are included in the bottom vertical level (e.g., burial, resuspension,
nitrification/denitrification, remineralization, sediment $O_2$ and $CO_2$ exchange). Light attenuation throughout the water column
is based on the diffuse attenuation coefficient (Kd), which is parameterized as a function of surface total suspended solids
(TSS; including inorganic and organic components) and salinity as a proxy for colored dissolved organic matter (Feng et al.,
2015; Turner et al., 2021). The sediment transport module within ECB is comprised of two vertical seabed layers that simulate
four suspended sediment size classes (Turner et al., 2021).



The carbon module within ECB has been fine-tuned in this implementation of the model, allowing for greater
performance in carbonate system simulations (Da et al., 2024). The model grid includes tidal wetlands along the York River
based on estimated wetland areas (Mitchell et al., 2017), which further contribute to TA fluxes through sulfate reduction in
sediments (Raymond et al., 2000; Najjar et al., 2020). $\Omega_{Ca}$ is calculated from DIC, TA, temperature, and salinity using CO2SYS
(van Heuven et al., 2011) using the equilibrium constants of Cai and Wang (1998) as they are suitable for both fresh and
estuarine waters (Dinauer and Mucci, 2017; Herrmann et al., 2020). Although submerged aquatic vegetation is a possible
source of $CaCO_3$ (Mazarrasa et al., 2015; Su et al., 2020), $CaCO_3$ precipitation and dissolution are not simulated in ECB due
to both insufficient observations and low submerged aquatic vegetation presence throughout the model domain (Orth et al.,
1998; Moore et al., 2009).
Several updates have been made in this implementation of ROMS-ECB to better represent oxygen and primary
production dynamics in the lower Virginia tributaries. The minimum phytoplankton growth rate has been increased to 2.15 d$^{-}$
$^{1}$, and the growth rate is limited in the fresh portion of the tributaries using a Michaelis-Menten function of salinity and a half-
saturation of 1.5 (Da et al., 2024). The sediment bed climatology from Moriarty et al. (2021) has been adjusted to better
represent the sand class distributions published in Nichols (1991) and observations taken by the USGS (Reid et al., 2005).
Specifically, the changes include a greater percentage of small clay-rich flocs throughout the main stem of the York River as
well as more sand and large silt-rich flocs in the Rappahannock River. Previously, the sediment module assumed the same
critical shear stress for large silt-rich flocs, small clay-rich flocs, and unaggregated mud; here, the critical shear stress of smaller
particles is lower than larger particles, meaning smaller particles resuspend more easily. The updated critical shear stress
coefficient for erosion and deposition is 0.5 Pa for large silt-rich flocs and 0.4 Pa for both small clay-rich flocs and
unaggregated mud, which represent a small portion of the sediment bed. The ballasting formulation of Turner et al. (2021) has
also been added to simulate the increase in particle sinking rates due to the aggregation of particles in turbid waters.

## 2.1.3 Oyster bioenergetics model

As part of this study, the oyster bioenergetics model EcoOyster (Brush and Kellogg, 2018; Kellogg et al., 2018) has
been one-way coupled to ROMS-ECB in the deepest (bottom) level (see Supplementary Tables S1-S4 for EcoOyster
equations). The one-way coupling allows the focus in this analysis to be on the effect of future climate change on oyster
growth, rather than the effect of oyster growth on water quality, which has been previously studied in the Chesapeake Bay
(e.g., Gawde et al., 2024). By focusing on the deepest vertical level of the model, the assumption is that oysters are growing
on the bottom, and not inside floating cages. The coupled model, referred to hereafter as ROMS-ECBO, simulates daily somatic
tissue dry weight, gonadal tissue dry weight, shell dry weight, and shell height of diploid and triploid oysters as a function of
filtration, respiration, egestion, allocation to reproduction, calcification, and dissolution (Brush and Kellogg, 2018; Kellogg et
al., 2018; Rivest et al., 2023). For the purpose of this study, only diploid oysters were included, as the triploid allometric
equations are not as well constrained. Tissue growth rates depend on individual weight together with temperature, salinity, $O_2$,
TSS, and particulate organic carbon (POC) from ROMS-ECB. Chl$a$ is required for the filtration function and is calculated



from ROMS-ECB phytoplankton carbon and Kd, in combination with seasonal carbon:chl*a* ratios that are computed using
equations from Cerco and Noel (2004). The calcification function includes a threshold value of $\Omega_{Ca} = 0.93$, determined through
laboratory experiments with Eastern oysters (Rivest et al., 2023).

The *EcoOyster* equations were developed from a meta-analysis of existing oyster bioenergetics models and laboratory

experiments with diploid oysters (Brush and Kellogg, 2018; Kellogg et al., 2018; Rivest et al., 2023). Allometric relationships
between shell dry weight, tissue dry weight, and shell height used for initial conditions were derived from observational data
in the Chesapeake Bay (VOSARA, 2024). Total dry tissue weight is calculated as the sum of somatic tissue weight and gonadal
weight. Reproduction is simulated through gonadal weight, a function of growth of gonadal tissue, resorption of gonadal tissue,
and spawning (Hofmann et al., 1994). Somatic tissue weight is a function of assimilation, respiration, growth of gonadal tissue,
and resorption of gonadal tissue. Assimilation is a function of filtration and POC. Filtration is a function of a maximum
filtration rate based on tissue weight, limited by sub-optimal temperature, salinity, TSS, $O_2$, and chl*a* (Cerco and Noel, 2005;
Fulford et al., 2007; Ehrich and Harris, 2015). The optimal temperature for oyster filtration ($T_{opt}$) is set to 27 °C (Jordan, 1987).
Filtration is also multiplied by *p*, a tunable factor representing the proportion of computed filtration actually performed by
oysters, which accounts for processes excluded from the model such as time spent filtering and is constrained by published
growth rates. Respiration is a function of tissue weight, temperature, and assimilation. While filtration has a temperature
limitation, respiration increases exponentially with temperature (Fig. S1). Tissue growth functions are not affected by
carbonate chemistry variables, as experimental studies have found that neither filtration (Lemasson et al., 2018) nor respiration
(Beniash et al., 2010; Matoo et al., 2013) of oysters are affected by pH changes; however, weight-specific net calcification is
a function of $\Omega_{Ca}$ and temperature (Rivest et al., 2023). Shell growth is a function of both total tissue weight and net
calcification.

**2.2 Present day reference simulation**

A realistic reference simulation was generated to represent 2017 conditions. The year 2017 was chosen for

atmospheric, terrestrial, and open-ocean boundary conditions as this represents a relatively typical hydrological year.
Atmospheric forcings (air temperature, long- and short-wave radiation, precipitation, winds, dewpoint temperature, and air
pressure) are obtained from the ERA5 atmospheric reanalysis (Copernicus Climate Change Service, 2017; Hersbach et al.,
2020). Surface atmospheric variables are available at 3-hourly intervals with a 0.25° resolution and are interpolated to a 0.2°
grid. Terrestrial inputs of freshwater, nitrogen, carbon, and sediment are derived from the Phase 6 CBP Watershed Model
(CBPWM; Bhatt et al., 2023) and USGS data. Daily estimates of freshwater discharge, water temperature, and loadings of
nitrate, ammonium, organic nitrogen, and four classes of sediment from the CBPWM were concatenated to 74 locations
throughout the model domain. To compute carbon loadings, constant carbon-to-nitrogen ratios are used, specifically 10:1 for
dissolved organic matter (Hopkinson et al., 1998) and 6.625:1 for particulate organic matter (Redfield, 1934). Riverine TA
concentrations are computed as in Da (2023), using monthly-varying linear relationships between historical USGS
observations of discharge and USGS TA estimates determined using the Weighted Regression on Time, Discharge, and Season



(WRTDS; Hirsch et al., 2010) approach. Riverine DIC is calculated from daily riverine TA and daily DIC:TA ratios, linearly
interpolated from the monthly climatology of USGS WRTDS DIC:TA in each tributary. As in Da (2023), open boundary
conditions are derived from a recent 600 m resolution whole-bay implementation of ROMS (St-Laurent and Friedrichs, 2024).
Initial conditions for the six-month spin-up were derived from previous model results (Da et al., 2024).
Since spring-spawned oysters are typically deployed in late spring through summer on oyster farms, the reference run
was started on July $1^{st}$ and spanned one full year, ending June $30^{th}$ of the following year. Oyster sizes were initialized based
on shell height approximations of a typical spring-spawned oyster at deployment in July (i.e., a few months old). Starting dry
tissue weight was assumed to be 0.001 g for all oysters, back-calculated from the approximate height of an oyster at the time
of deployment. Starting shell dry weights and heights were calculated from allometric relationships to be 0.144 g and 11.6
mm, respectively.

### 2.3 Comparison of reference simulation to in situ observations

*In situ* water quality monitoring observations are available since 1984 throughout the Chesapeake Bay. Specifically,
the Chesapeake Bay Program's Water Quality Monitoring Program (CBP WQMP) conducts cruises in the Bay and its
tributaries. On average, stations are sampled once monthly, with the exception of June through August in the mainstem, when
sampling occurs twice. In this study, measurements of water temperature, salinity, $O_2$, pH (NBS scale), TSS, and POC are
used from 16 CBP stations throughout the model domain, with depths ranging from 5 to 16 m (Fig. 1a; CBP, 2024). For all
variables except TSS and POC, measurements are taken *in situ* using a YSI or Hydrolab® sonde roughly every one to two
meters of the water column. TSS and POC are obtained from bottle samples at the surface, bottom, and at deeper stations, two
additional depths above and below the pycnocline. TSS is determined by filtering a known volume of water through a pre-
weighted filter and then re-weighing the filter after filtration and drying. POC is determined through combustion of a filter
using an elemental analyzer (Olsen, 2012).
Model skill was evaluated by comparing results from the reference simulation to the CBP WQMP observations
described above. Hourly outputs from the four closest grid cells to each CBP station were spatially interpolated to obtain results
at each respective station. Multiple variables in ECB at the bottom level of the model, including temperature, salinity, $O_2$, pH,
TSS, and POC, were compared with observations from the same station and time, within the bottom 10% of the water column
(Table 1). Model bias and root-mean squared difference (RMSD) were computed for all aforementioned variables. Seasonal
skill was also evaluated by comparing the 2017 reference run to CBP decadal averages (Figs. 2, 3). Decadal means were used
for these comparisons, as once-monthly or once-seasonally sampling dates in 2017 bias outputs toward conditions on the time
of the month when the measurements were taken in 2017, and the purpose of the comparison was to examine how the model
reproduces average seasonal variability.










**Figure 2. Seasonally-averaged bottom (a) temperature, (b) salinity, (c) dissolved oxygen, and (d) pH from the reference run. Circles represent seasonal decadal-averaged *in situ* observations at Chesapeake Bay Program stations (2010-2020). (DJF = winter, MAM = spring, JJA = summer, and SON = fall). Figure 1 ROMS-ECBO model domain of Chesapeake Bay tributaries illustrating (a) bathymetry in meters and locations of Chesapeake Bay Program water quality monitoring stations (red circles) and (b) bottom salinity zones.**

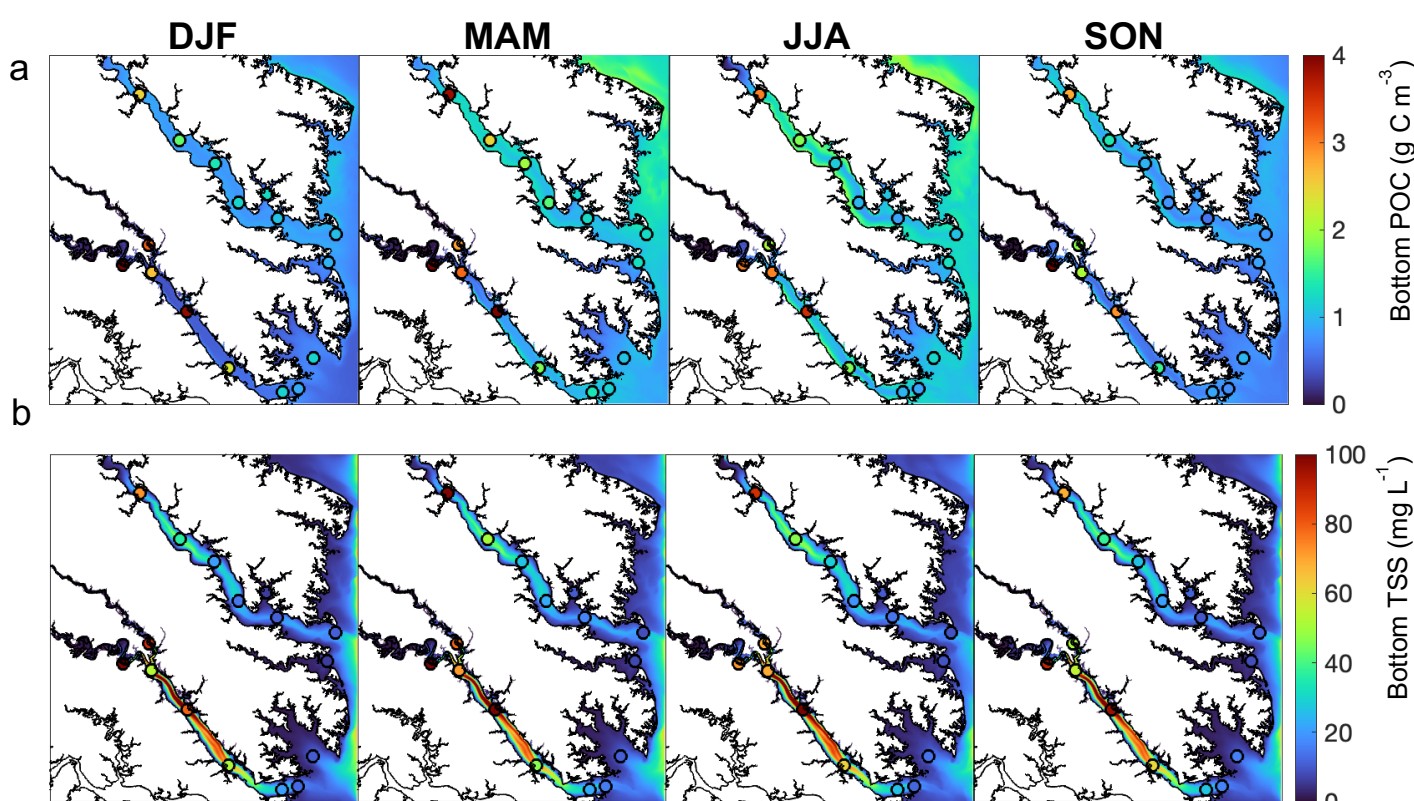

**Figure 3. Seasonally-averaged bottom (a) POC and (b) TSS from output of the reference run of the ROMS-ECBO model. Circles represent seasonal decadal-averaged bottom measurements at Chesapeake Bay Program stations (2010-2020). (DJF = winter, MAM = spring, JJA = summer, and SON = fall).**

When compared to 2017 WQMP observations and seasonal decadal averages, model skill of ROMS-ECBO is reasonably high (Table 1, Figs. 2, 3), and similar to other model implementations of the Chesapeake Bay (Irby et al., 2016). Temperature and salinity are reproduced relatively well year-round (Fig. 2a,b) with annual biases of only 0.2°C and -1.5, respectively (Table 1). Bottom $O_2$ and pH are slightly overestimated, exhibiting the greatest model-data misfit in the spring and summer months in the tributary channels (Fig. 2c,d). pH is overestimated by 0.2 units, which is within the accuracy of the electrode measurements. Observed POC concentrations in the York and upper Rappahannock are higher than simulated in the



model and exhibit very high spatial variability (Fig. 3a). Despite the high spatial variability of the TSS observations (Fig. 3b),
mean TSS ($45 \pm 54$ mg L$^{-1}$) is captured within 1.1 mg L$^{-1}$ by the model.
Growth rates determined using the *EcoOyster* equations and environmental outputs from ROMS-ECB were compared
with oyster data collected in the York River (Paynter et al., 2008; Liddel, 2008; Kingsley-Smith et al., 2009; Degremont et al.,
2012; Callam et al., 2016). Specifically, the tunable parameter ($p$) that limits oyster filtration was adjusted to provide a best
match between the modeled oyster growth rates and the published rates. Multiple $p$-values were tested, and a value of $p=0.15$
resulted in modeled oyster growth that best matched published growth rates. The resulting shell growth predicted by the model
was found to be close to the *in situ* data ($52.0 \pm 1.1$ mm y$^{-1}$ and $51.3 \pm 2.9$ mm y$^{-1}$ for the model and observation means and
standard deviations, respectively).
**Table 1. Model skill statistics (mean ± standard deviation) comparing bottom grid cells from the reference run to Chesapeake Bay**
**Program observations from the same station location and time, within the bottom 10% of the water column.**

| Variable | Model | Observation | Model Bias | RMSD[a] |
|---|---|---|---|---|
| **Temperature (°C)** <br> **n = 130** | $17.0 \pm 9$ | $16.7 \pm 9$ | + 0.2 | 0.7 |
| **Salinity** <br> **n = 127** | $13.9 \pm 7$ | $15.4 \pm 7$ | -1.5 | 2.7 |
| **Oxygen (mg O$_2$ L$^{-1}$)** <br> **n = 130** | $8.0 \pm 2.3$ | $7.2 \pm 2.9$ | +0.9 | 1.3 |
| **pH** <br> **n = 74** | $7.8 \pm 0.4$ | $7.6 \pm 0.4$ | + 0.2 | 0.4 |
| **TSS (mg L$^{-1}$)** <br> **n = 74** | $44 \pm 34$ | $45 \pm 54$ | -1.1 | 48.3 |
| **POC (g C m$^{-3}$)** <br> **n = 74** | $0.7 \pm 0.3$ | $1.7 \pm 2.1$ | -1.0 | 2.4 |

[a]RMSD = root mean squared difference
**2.4 Future simulations**
In addition to the reference run, this study generated five future simulations (Table 2) to investigate the change in
carbonate chemistry conditions and oyster growth resulting from three drivers of future change in the bay: increased
atmospheric CO$_2$ (*AtmCO$_2$*), atmospheric warming (*Temp*), and reduced nutrient loading (*TMDL*). Model forcings were
modified for each simulation to represent mid-century conditions. A *Combined Future* simulation was run including forcings
of all future stressors, in addition to three sensitivity simulations to isolate the impacts of each stressor on oyster growth.
Atmospheric CO$_2$ concentration for the *AtmCO$_2$* and *Combined Future* simulations was set to 655 ppm, derived from the
Coupled Model Intercomparison Project Phase 5 report RCP8.5 (business-as-usual) scenario projected for 50 years in the



future relative to the reference run (Meinshausen et al., 2011). Future atmospheric temperature for the *Temp* and *Combined Future* simulations was obtained from the IPSL-CM5A-LR Earth System Model (Dufresne et al., 2013), statistically downscaled with the Multivariate Adaptive Constructed Analogs method (Abatzoglou and Brown, 2012). IPSL-CM5A-LR was selected as in Hinson et al. (2023), since it was deemed the most representative downscaled ESM of the 20 available. As in Hinson et al. (2023), the delta method was used to calculate the daily average change in atmospheric temperatures between present-day and future conditions. To calculate this change, two 30-year climatologies, centered on 2000 and 2050 respectively, were computed and daily averaged 50-year differences between the two climatologies (Fig. 4) were added to the atmospheric temperatures used in the reference run. Future watershed inputs for the *TMDL* and *Combined Future* simulations included a climatology of nitrate, ammonium, dissolved organic matter, and particulate organic matter concentrations, derived from a Phase 6 CBPWM 1991-2000 run using reduced nutrient concentrations assuming the TMDLs had been successfully achieved (Bhatt et al., 2023). Freshwater discharge in this run was set to be identical to the reference run, to isolate the effects of lowered nutrient concentrations on water chemistry and oyster growth. Since future climate change is expected to impact terrestrial inputs much less than future management actions (Irby et al., 2018), the direct impact of climate change on the watershed is not considered in this analysis. A fifth simulation (*AtmCO2 + Temp)* was run to compare the influences of local management actions to the combined drivers of climate change, which includes both future atmospheric $CO_2$ concentration and atmospheric temperature. Preliminary investigations revealed a minimal impact of sea level rise on $\Omega_{Ca}$ in the bay; therefore, it was not included in the simulated climate change variables.

**Table 2. Experimental design for future simulations conducted for comparison to reference run. Model forcings include a combination of 2017 (reference) and 2067 (future) inputs of atmospheric CO2, atmospheric temperature, and terrestrial nutrient loadings.**

| Future Simulation Name | Atmospheric CO₂ | Atmospheric Temperature | Terrestrial inputs |
|---|---|---|---|
| **Combined Future** | Future | Future | TMDL[a] |
| **AtmCO₂** | Future | Reference | Reference |
| **Temp** | Reference | Future | Reference |
| **TMDL** | Reference | Reference | TMDL |
| **AtmCO₂ + Temp** | Future | Future | Reference |

[a]TMDL (Total Maximum Daily Load) forcing includes inputs of nitrate, ammonium, and dissolved and particulate organic matter under the assumption that the nutrient reduction goals (EPA, 2010) are met.



**Figure 4. Monthly-averaged 50-year atmospheric temperature differences over the ROMS-ECBO model domain calculated as projections from 2050 minus those from 2000.**



To generate open boundary conditions for each future simulation, a full bay model (St-Laurent and Friedrichs, 2024) was run with the same atmospheric and river forcings as in this 120-meter model implementation. As in the reference run, all future simulations were spun up for six months (January 1 – June 30) before beginning on July 1, but represent 50 years in the future from the reference simulation (i.e., July 1, 2067). Initial conditions for all spin-ups are identical to the reference simulation. Analysis confirmed the effects of initial conditions are negligible by July 1. To examine results most relevant to oysters, model output was extracted at locations that support oyster production, defined as all grid cells in which tissue weight exceeded 1 g at the end of the reference run (i.e., one year of growth; Fig. S2). All results shown are from the bottom level of the model, representing conditions similar to on-bottom or bottom cage aquaculture methods that are common in Virginia. Spatial variation in model outputs across grid cells in the model domain is reported using standard deviation.

## 3 Results

### 3.1 Reference run results

In the present-day reference run, the environmental variables used as inputs to the oyster parameterizations exhibit substantial seasonal (Fig. 5a-f) and spatial (Figs. 6, S3) variability. As expected, bottom temperature is highest in summer, reaching an average of 29.3 °C in July when averaged across grid cells that support oyster growth (Fig. 5a). Temperature is higher in the shallower parts of the tributaries compared to the channels (Fig. S3a). Bottom salinity exhibits higher values in the fall and winter, reaching a maximum average of 17.7 in October, and drops in the spring and summer to reach a minimum average of 12.3 in June (Fig. 5b). Annual average bottom salinity ranges from 0 to 26 throughout the model domain (Fig. S3b), with the highest values in the southern areas in closest proximity to the open-ocean. The seasonal cycle for bottom POC is similar to that of temperature, peaking at 1.7 g C m$^{-3}$ in June and dropping to 0.57 g C m$^{-3}$ in January (Fig. 5c). Bottom POC also varies widely throughout the model domain (Fig. 6a), with relatively higher values in the Rappahannock compared to the York River, along the shoals of the tributaries, and along the western shoals of the mainstem Bay north of the Rappahannock. $\Omega_{Ca}$ exhibits an annual cycle similar to that of temperature and POC, reaching a maximum average of 3.2 in August and a minimum average of 1.1 in January. Annual mean bottom $\Omega_{Ca}$ also varies widely throughout the model domain (Fig. 6d). Generally, bottom $\Omega_{Ca}$ increases with salinity, with low to zero values in the tidal fresh portions of the upper tributaries and higher values along the western shoals of the mainstem Chesapeake Bay. The opposite temporal pattern is seen in bottom $O_2$, which peaks at 12.3 mg L$^{-1}$ in February and drops to an average of 6.3 mg L$^{-1}$ in August (Fig. 5e). $O_2$ concentrations are highest along the shoals and lowest in the deep channels (Fig. S3c). Bottom TSS concentrations exhibit tidal variability throughout the year and are highest in the York River with much lower concentrations observed in the other portions of the model domain (Fig. S3d). Environmental conditions averaged annually across grid cells that support oyster growth are provided in Table 3, and conditions averaged annually across all grid cells in the model domain are provided in Table S5.




**Figure 5. Time series of daily bottom (a) temperature, (b) salinity, c) POC, (d) ΩCa, (e) shell weight, and (f) tissue weight, averaged over grid cells that support oyster growth in the reference run, for the present-day reference run (black line) and Combined Future simulation (blue line).**





**Figure 6. Annual mean bottom (a-c) POC and (d-f) $\Omega_{Ca}$ from (a,d) the present-day reference run, (b,e) the *Combined Future* simulation, and (c,f) the difference between (a) and (b), i.e., *Combined Future* minus reference.**


Tissue and shell weights increase modestly from July through April, and the highest rates of increase are seen in May and June near the end of the one-year reference run (Fig. 5g,h). At the end of the reference run, the spatial patterns of shell and tissue weight are nearly identical (Fig. 7), as tissue growth largely drives shell growth (Table S4). Both shell and tissue weights are highest along the shoals of the York and Rappahannock Rivers (Fig. 7a,d) and low in the deeper waters where TSS concentrations are high (Fig. S3d). A wider region of high shell and tissue weight appears in the Rappahannock, while the highest weights in the York are confined to a very narrow and shallow strip along the coastline. Shell and tissue weights are higher along the southwestern than the northeastern coastlines of the tributaries, where the shoals are wider in both tributaries (Fig. 1a). Oyster growth metrics averaged across grid cells that support oyster growth are provided in Table 4.



**Table 3. Bottom environmental variables for each model simulation (annual mean ± standard deviation) for grid cells that support oyster growth in the reference run (defined as those with greater than 1g dry tissue weight after one year of growth; Fig. S2). Analogous results averaged over all model grid cells are shown in Table S5.**

| Model Simulation | Temperature (°C) | Salinity | POC (g C m$^{-3}$) | $\Omega_{Ca}$ | Dissolved Oxygen (mg O$_2$ L$^{-1}$) | TSS (mg L$^{-1}$) |
|---|---|---|---|---|---|---|
| Reference | 17.0 ± 0.7 | 15.7 ± 2.1 | 1.12 ± 0.1 | 2.5 ± 0.49 | 9.1 ± 0.6 | 11.4 ± 5.8 |
| Combined Future | 18.5 ± 0.8 | 16.0 ± 2.1 | 1.03 ± 0.1 | 1.6 ± 0.35 | 8.7 ± 0.6 | 11.1 ± 5.9 |
| AtmCO$_2$ | 17.0 ± 0.7 | 15.7 ± 2.1 | 1.12 ± 0.1 | 1.6 ± 0.35 | 9.1 ± 0.6 | 11.4 ± 5.8 |
| Temp | 18.5 ± 0.8 | 16.0 ± 2.1 | 1.07 ± 0.1 | 2.5 ± 0.41 | 8.8 ± 0.6 | 11.1 ± 5.9 |
| TMDL | 17.0 ± 0.7 | 15.7 ± 2.1 | 1.08 ± 0.1 | 2.4 ± 0.53 | 9.1 ± 0.6 | 11.2 ± 5.9 |
| Temp + CO$_2$ | 18.5 ± 0.8 | 16.0 ± 2.1 | 1.07 ± 0.1 | 1.7 ± 0.33 | 8.8 ± 0.6 | 11.1 ± 5.9 |

**Table 4. Modeled oyster characteristics from the end of each simulation (mean ± standard deviation) over grid cells that support oyster growth in the reference run (defined as those with greater than 1g dry tissue weight after one year of growth; Fig. S2).**

| Model Simulation | Shell Weight (g) | Tissue Weight (g) | Shell Thickness (g mm$^{-1}$) |
|---|---|---|---|
| Reference | 16.8 ± 10.9 | 2.2 ± 1.5 | 0.18 ± 0.08 |
| Combined Future | 5.4 ± 5.7 | 0.9 ± 0.8 | 0.07 ± 0.05 |
| AtmCO$_2$ | 10.5 ± 8.0 | 2.2 ± 1.5 | 0.12 ± 0.06 |
| Temp | 9.7 ± 9.1 | 1.2 ± 1.1 | 0.10 ± 0.07 |
| TMDL | 13.1 ± 8.2 | 1.7 ± 1.2 | 0.15 ± 0.06 |
| Temp + CO$_2$ | 6.6 ± 7.1 | 1.2 ± 1.1 | 0.08 ± 0.06 |

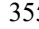

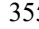

**Figure 7. (a-c) Shell weight and (d-f) tissue weight at the end of the one-year simulation from (a,d) the present-day reference run, (b,e) the *Combined Future* run, and (c,f) their difference, i.e., *Combined Future* minus reference.**

## 3.2 Results of Combined Future simulation

All environmental variables examined exhibit change from the reference run in the *Combined Future* simulation. Temperature and salinity are projected to increase across the entire model domain (Fig. S3a,b). When averaged over the model domain, temperature is projected to increase by $1.5 \pm 0.26$°C, and salinity is projected to increase by $0.21 \pm 0.11$ (Table S5). Bottom POC is projected to decrease by $0.07 \pm 0.05$ g C m$^3$ (Table S5), with POC reductions predicted to be most pronounced in the mid- to upper tributaries (Fig. 6c). Mid-century bottom $\Omega_{Ca}$ is projected to be lower throughout most of the region, with an average reduction of $0.8 \pm 0.19$ over the whole model domain (Table S5). The spatial distribution of future $\Omega_{Ca}$ is generally consistent with present-day $\Omega_{Ca}$ patterns, and the greatest decreases are projected to occur in regions with the highest present-



day $\Omega_{ca}$ (Fig. 6 d,e,f). An average reduction in $O_2$ of $0.3 \pm 0.08$ mg L$^{-1}$ is predicted across the model domain (Table S5), which
will be mostly spatially uniform (Fig. S3c). TSS is projected to be reduced by $0.20 \pm 0.25$ mg L$^{-1}$ with high spatial variability
in the projected change (Fig. S3d).

Changes in environmental conditions do not occur uniformly throughout the year. Temporal changes in environmental

conditions averaged across grid cells that support oyster growth are provided in Figure 5. Annually averaged increases in
temperature and salinity are the same when averaged over only grid cells that support oyster growth as they are when averaged
across the entire model domain (Tables 3, S5). The greatest temperature increases are projected to occur in the warmer months,
with an average increase of 1.6 °C predicted for June through August and an average increase of 1.2 °C predicted for December
through February. Bottom temperatures are projected to surpass the optimal temperature for oyster filtration (27 °C) primarily
in July and August (Fig. 5a). Salinity increases are projected to be greatest at the beginning of the year, with an average increase
of 0.44 between January and March and an average increase of 0.20 for the remainder of the year (Fig. 5b). Bottom POC at
grid cells that support oyster growth is expected to decrease slightly less than the average for the entire region (Tables 3, S5),
with the greatest reductions in the spring and summer and little to no change in the winter (Fig. 5c). For $\Omega_{Ca}$, $O_2$, and TSS,
projected reductions are slightly greater at oyster growth sites than for the entire domain. $\Omega_{Ca}$ is projected to decrease by 0.9,
with the greatest reductions expected to occur the warmer months (Fig. 5d). $O_2$ is projected to decrease year-round, though
with slightly greater reductions in the winter (Fig. 5e) and an annual average reduction of 0.4 mg $O_2$ L$^{-1}$ (Table 3). TSS is
projected to decrease annually by 0.3 mg L$^{-1}$ (Table 3), mostly in the spring, due to lowered POC (Fig. 5f).

Modeled shell and tissue weights after one year of growth are projected to decline in all regions that exhibit present-

day growth, with the most severe reductions (up to 100%) occurring along the York and Rappahannock River shoals (Figs.
7c,f, 8). One-year tissue weight will be reduced by 1.3 g, on average, representing a 60% reduction across grid cells that support
oyster growth (Table 4). Shell weight, which is largely driven by changes in tissue weight, is projected to be reduced by 11.4
g on average after one year of growth, representing a 68% reduction in average shell weight in regions that support oyster
growth (Table 4). The greatest reduction in shell and tissue growth rates will occur in the warmer months near the end of the
one-year simulation (-0.1 g d$^{-1}$ from May through June), whereas the smallest change will occur in the winter months (-0.02 g
d$^{-1}$ from December through February), as the least growth occurs during that time (Fig. 5g,h). Shell thickness, calculated as
the ratio of shell weight to shell height, will be reduced by 61% on average (0.11 g mm$^{-1}$; Table 4).

Declines in year one shell weight will vary throughout the model domain (Fig. 8), following relative changes in

bottom POC and $\Omega_{Ca}$ (Fig. 9). The mainstem has the most moderate reduction in shell weight relative to reference shell weight,
with an average reduction of 31%, indicated by the slope of the scatterplot. Shell weights in the Rappahannock and York face
the steepest reductions relative to reference, with average reductions of 86% and 96%, respectively, and a large portion of
York oysters facing complete depletion of oyster tissue and shell in these locations (Fig. 9; indicated by proximity to 1:1 line).
Proportional shell weight reductions in the mainstem are projected to correlate with POC reductions (Fig. 9a). For $\Omega_{Ca}$ in the
mainstem, a group of sites face the greatest proportional reductions when $\Omega_{Ca}$ reductions are the greatest. However, for sites
with lower proportional shell loss, the opposite trend is observed (Fig 9d). In the Rappahannock, higher POC reductions

coincide with slightly lower proportional shell loss (Fig. 9b). Sites with the largest reductions in POC primarily occur in the York (Fig. 9c; see dark blue symbols on the 1:1 line), and the greatest proportional shell weight reductions coincide with the greatest POC and $\Omega_{Ca}$ reductions (Fig. 9c,f). Similar results are found for tissue weight (not shown).

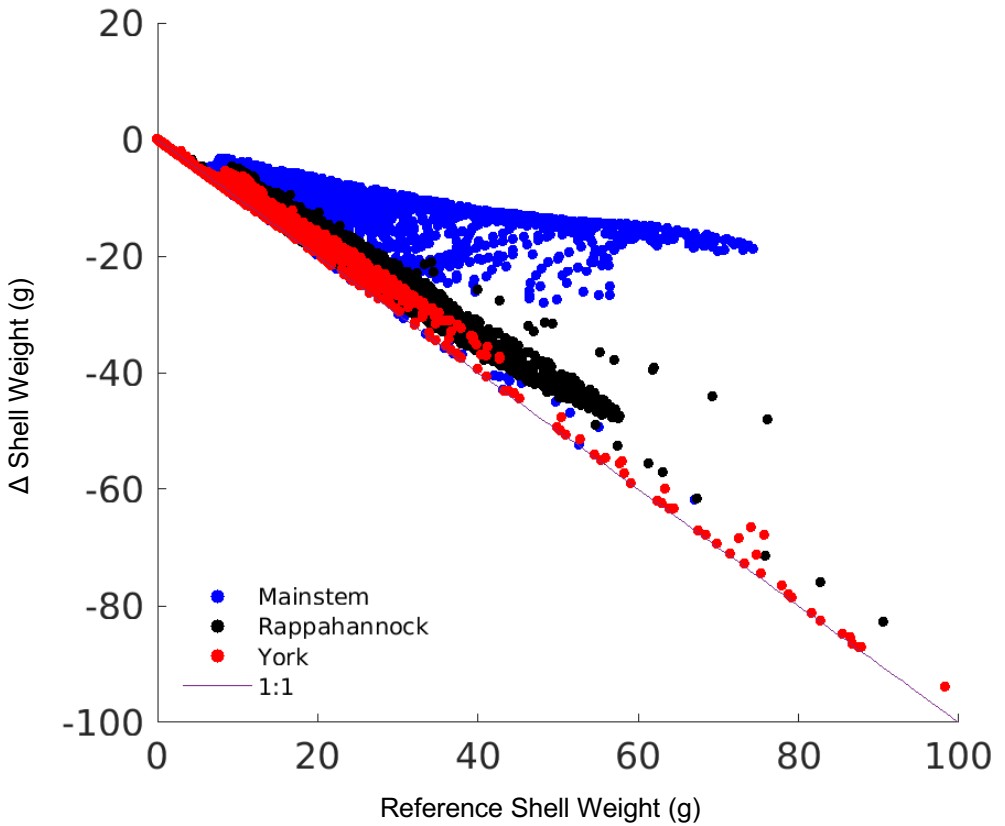

**Figure 8. Difference in shell weight at the end of the one-year simulation between the *Combined Future* run and the reference run, i.e., *Combined Future* minus reference, colored by region. Each point represents a grid cell where oyster growth occurs in the reference run.**





**Figure 9. Difference in shell weight at the end of the one-year simulation between the *Combined Future* run and the reference run colored by (a-c) change in POC and (d-f) change in bottom $\Omega_{Ca}$ (i.e. *Combined Future* minus reference) for grid cells that support oyster growth. Results are presented for (a,d) the mainstem shoal only, (b,e) the Rappahannock River only, and (c,f) the York River only.**

### 3.3 Results of individual future sensitivity simulations

Four individual future sensitivity simulations were conducted to isolate the specific mechanisms (increased atmospheric CO₂, increased atmospheric temperature, and/or nutrient reductions) causing the projected changes described above in the *Combined Future* simulation. The *AtmCO2* sensitivity simulation produces substantial reductions in average bottom $\Omega_{Ca}$ (Fig. 10d) and, as expected, is not projected to substantially impact bottom temperature, salinity, POC, O₂, or TSS (Table 3; Figs. 10a, S4). The projected reduction in $\Omega_{Ca}$ is 0.9 when averaged over oyster growth sites (Table 3), 0.1 greater in magnitude than the average reduction for the entire model domain (Table S5), as greater reductions are expected along the shoals of the Rappahannock and mainstem shoal than the York and upper section of the Rappahannock (Fig. 10d). In this





*AtmCO₂* simulation, shell weight is predicted to be most steeply reduced in the Rappahannock, with less impact in the York
and mainstem regions (Fig. 11a). At grid cells with oyster growth, *AtmCO₂* produces a shell weight reduction of 6.3 g in
comparison to the reference simulation, but no change in tissue weight (Table 4).

**Figure 10. Differences in annual averaged (a-c) bottom POC and (d-f) bottom $\Omega_{Ca}$ (d-f) for three sensitivity experiments: (a)**
**AtmCO₂, (b) Temp, and (c) TMDL. Differences represent future results minus those from the present-day reference run.**





**Figure 11 Differences in (a-c) shell weight and (d-f) tissue weight at the end of the one-year simulation for three sensitivity experiments: (a) AtmCO₂, (b) Temp, and (c) TMDL. Differences represent future results minus those from the present-day reference run.**

The *Temp* sensitivity simulation produces changes in all environmental variables impacting oyster growth, with the exception of $\Omega_{Ca}$ (Tables 3, S5). Average changes in temperature, salinity, and TSS will be identical to those from the *Combined Future* simulation (Tables S5, 3). Predicted reductions in POC and $O_2$ will be smaller in magnitude than in *Combined Future,* though more severe than any other single sensitivity experiment (Table 3). Temperature and salinity will increase across the entire model domain, with a greater salinity increase occurring in the Rappahannock and along the mainstem shoal (Fig. S4a-c). TSS will decrease primarily in the channels of the lower York and Rappahannock and on the mainstem shoal (Fig. S4d). POC reductions are expected to cover the majority of the model domain, with larger reductions in the upper Rappahannock (Fig. 10b). Slight increases in $\Omega_{Ca}$ are observed in shallow tidal creeks (Fig. 10e); however, no substantial





change in average $\Omega_{Ca}$ is predicted (Table 3, S5). $O_2$ at oyster grid cells will exhibit a similar but slightly smaller average
reduction compared to *Combined Future* (Table 3). Patterns of change in shell weight in the *Temp* sensitivity simulation
resemble those in the *AtmCO₂* simulation (Fig. 11b), with additional reductions along the mainstem shoal and a greater
predicted mean reduction of 7.1 g, a 42% decrease at grid cells with oyster growth (Table 4). Unlike *AtmCO₂*, tissue weight
will decrease in *Temp*, by an average of 1 g, a 46% reduction (Table 4).
The *TMDL* sensitivity simulation produces a much smaller average change in environmental conditions than the
*AtmCO₂* or *Temp* simulations (Tables S5, 3). *TMDL* does not substantially influence temperature, salinity, or $O_2$ (Tables S5,
3, Fig. S4), but produces POC and TSS reductions close to the averages for *Temp* (Tables S5, 3). While POC change in the
*Temp* simulation is concentrated in the deeper portions of the tributaries (Fig. 10b), the POC reductions in the *TMDL* simulation
are concentrated along the shoals of the tributaries, with the greatest reductions in the upper Rappahannock (Fig. 10c). TSS
changes in *TMDL* are limited to the tributaries, occurring along the shoals of the Rappahannock and in patches throughout the
York (Fig. S4d). Future change in $\Omega_{Ca}$ in this simulation is less than for *AtmCO₂* and is largely confined to the upper
Rappahannock shoals and in shallow tidal creeks throughout the study region (Fig. 10f). Patterns of change in shell weight
will resemble *AtmCO₂* and *Temp* in the tributaries, but no change is predicted along the mainstem shoal (Fig. 11c). The *TMDL*
simulation produces reduced shell (3.7 g) and tissue (0.5 g) weights, with a smaller negative influence on shell and tissue
weight than *Temp* (Table 4; Fig. 11c, d).
Environmental conditions in the *AtmCO₂ + Temp* simulation are nearly identical to those in the *Combined Future*
simulation (Tables 3, S5), with the exception of $\Omega_{Ca}$, which is slightly higher due to the absence of *TMDL*'s influence. As tissue
growth is unaffected by $\Omega_{Ca}$, tissue weight in this simulation is identical to that of the *Temp* simulation. Average shell weight
reduction in *AtmCO₂ + Temp* is 10.2 g, slightly greater than from *AtmCO₂* alone, due to the combined influences of lowered
tissue growth and lower $\Omega_{Ca}$.
**4 Discussion**
This study provides high-resolution projections for oyster growing conditions and corresponding oyster growth in the
Chesapeake Bay, with a specific focus on two Virginia tributaries. A high-resolution hydrodynamic-biogeochemical model
was coupled with an Eastern oyster bioenergetics model and forced with future projections for atmospheric $CO_2$, temperature,
and nutrient management. An overall reduction in $\Omega_{Ca}$ and oyster growth are predicted by mid-century throughout the study
region under the combined effects of all three future stressors. Specifically, the greatest reductions in oyster growth are
projected to occur in the York and Rappahannock Rivers, where unfavorable conditions for calcification will expand in the
future and where food availability will be strongly impacted by warming and nutrient reductions. Bottom conditions in the
York and Rappahannock rivers, particularly in the upper portions, will likely be unsuitable for aquaculture at mid-century on
average, indicating climate change preparedness is critical for the oyster aquaculture industry.



## 4.1 Future projections of $\Omega_{Ca}$

The magnitude of future change in $\Omega_{Ca}$ varies with present-day $\Omega_{Ca}$ conditions. Regions with high present-day $\Omega_{Ca}$, primarily the mainstem shoals, are projected to experience the greatest reductions because of their low partial pressure of $CO_2$ ($pCO_2$) relative to fresher waters. Biologically driven low $pCO_2$ water on mainstem shoals has a greater capacity for $CO_2$ uptake from the atmosphere than high $pCO_2$ water, which is causing the fresher tributaries to experience smaller increases in DIC and smaller reductions in $\Omega_{Ca}$. Acidic freshwater input often causes $pCO_2$ in the upper tributaries to exceed atmospheric $pCO_2$, causing outgassing (Cai et al., 2017; Shen et al., 2019b; St-Laurent et al., 2020; Cai et al., 2021). Despite the lower Rappahannock having a lower salinity than the lower York, it also has a lower DIC to TA ratio, so the rate at which the lower Rappahannock absorbs $pCO_2$ is higher (Da et al., 2021). Total alkalinity, or buffering capacity, is also lower in the lower Rappahannock than the lower York, so the lower Rappahannock cannot resist changes in carbonate chemistry to the same degree as the York. As a result, we observe the Rappahannock changing faster than the York. Since higher $\Omega_{Ca}$ regions will experience greater reductions than lower $\Omega_{Ca}$ regions (Fig. 5), the overall spatial variability of $\Omega_{Ca}$ will be reduced by mid-century, and more areas will experience conditions that are unfavorable for oyster shell-building.

Although future atmospheric $CO_2$ and reduced nutrient loading will both contribute to $\Omega_{Ca}$ reductions, the modeling experiments conducted here highlight that increasing atmospheric $CO_2$ is the largest contributor to decreases in $\Omega_{Ca}$ throughout the study region. Increased atmospheric $CO_2$ will cause reductions in $\Omega_{Ca}$ across the model domain, while nutrient reductions are expected to mainly influence $\Omega_{Ca}$ in shallow and fresh coastal areas, with little influence in oyster growing regions. The effects of warming on $\Omega_{Ca}$ may slightly offset the influence of atmospheric $CO_2$ in certain areas, but this will likely only occur in fresh tidal creeks where oysters do not currently reside (Fig. 10e). Given the importance of atmospheric $CO_2$ in shaping future $\Omega_{Ca}$ conditions in the lower bay, reductions in anthropogenic carbon emissions will be necessary to lessen the projected impacts on carbonate chemistry in the Chesapeake Bay and globally.

Comparing our results to other studies examining the effects of acidification reveals that the Chesapeake Bay will likely acidify faster than the US West Coast. Siedlecki et al. (2021a) projected a decrease of 0.8-1.0 in $\Omega_{Ca}$ in the Northern California Current System between 2000 and 2100. Projections from the present work indicate a similar magnitude of reduction in the lower Chesapeake Bay over a shorter time period (50 years), suggesting a faster rate of acidification in the lower bay. Feely et al. (2009) also reported that projections for $\Omega_{Ca}$ reductions are slightly greater in the Atlantic than in the Pacific. The relative differences in rates of acidification should be considered, however, in the context of present-day $\Omega_{Ca}$. The Pacific Ocean has a higher ratio of DIC:TA than the Atlantic, so present-day Pacific $\Omega_{Ca}$ is lower (Feely et al., 2004; Dunne et al., 2012). Therefore, while the Chesapeake Bay is acidifying faster, coastal Pacific waters may become undersaturated with calcite and aragonite sooner than in Chesapeake Bay. US West Coast shellfish mortality events associated with acidification or other climate change stressors may place increased pressure on US Atlantic fisheries to provide shellfish to the nation, highlighting the importance of climate change preparedness and resilience in the Chesapeake Bay region.



500 While atmospheric $CO_2$ is primarily responsible for changes in $\Omega_{Ca}$, nutrient reductions are also projected to worsen
501 carbonate chemistry conditions. Eutrophication can suppress acidification by increasing primary production (Borges and
502 Gypens, 2010; Shen et al., 2019; Da et al., 2021), and when simulating a reduction in eutrophication via nutrient management
503 in our modeling study, the countering effect occurred. While the reduction in $\Omega_{Ca}$ from nutrient management is minor compared
504 to the projected impacts of $CO_2$-driven acidification, its small contribution may shift $\Omega_{Ca}$ conditions from favoring net
505 calcification to favoring net dissolution, demonstrating the importance of considering multiple drivers when predicting
506 exposure to ecologically relevant conditions of coastal acidification.

### 4.2 Future projections of oyster growth

508 Acidification, warming, and nutrient reductions are projected to affect shell and tissue growth of oysters in different
509 ways. Here, increased atmospheric $CO_2$ caused reductions in shell growth of Eastern oysters due to its negative effect on $\Omega_{Ca}$
510 and thus calcification rates, which is consistent with experimental studies (Waldbusser et al., 2011; Gobler and Talmage, 2014;
511 Himes et al., 2024). Shell weight reductions from increased atmospheric $CO_2$ were driven by changes in calcification rate
512 alone, as tissue weight in *EcoOyster* is unaffected by $\Omega_{Ca}$ (Fig. 11d; Rivest et al., 2023). Experimental studies have identified
513 indirect physiological impacts of elevated $CO_2$ on juvenile/adult oyster metabolism, growth, and reproduction (Beniash et al.,
514 2010; Dickinson et al., 2012), suggesting that increased atmospheric $CO_2$ can sometimes influence tissue growth. Further
515 investigation is necessary in order to include the relationship between atmospheric $CO_2$ and oyster tissue growth in *EcoOyster*.
516 Biological and chemical reactions occur faster at higher temperatures, meaning calcification rates may be higher under future
517 warming conditions (Waldbusser et al., 2011), as long as $\Omega_{Ca}$ is still high enough to support calcification. Conversely, under
518 conditions of extreme low $\Omega_{Ca}$, warming may exacerbate dissolution rates and shell weight reductions. Our results also show
519 that nutrient reductions will lead to reductions in shell weight, largely driven by a reduction in tissue weight resulting from
520 lower food availability (POC), rather than lower $\Omega_{Ca}$.

521 While nutrient reductions are projected to have little influence on $\Omega_{Ca}$ in this study, their negative influence on food
522 availability may be detrimental to tissue growth in certain parts of the study region, particularly the York River. Our model
523 projections suggest that nutrient reductions may in some cases produce conditions that do not support any oyster growth along
524 the shoals of the York (Fig. 9c; 10c), a result of reductions in food availability that are predicted to be more substantial in the
525 tributaries than the mainstem region (Fig. 6c). Multiple studies have demonstrated that Eastern oysters and other calcifying
526 organisms perform better under acidification when they have sufficient food availability, as they are better able to keep up
527 with the energetic demands of environmental stress (Thomsen et al., 2015; Ramajo et al., 2016; Schwaner et al., 2023).
528 Therefore, nutrient reductions will likely influence oyster growth under acidification stress by different magnitudes in each
529 tributary. When comparing the effects of local management actions to reduce nutrient runoff to the effects of climate change
530 (increased atmospheric $CO_2$ and warming), it is evident that, on average, climate change will have a much greater negative
531 influence on oyster growth (Table 4). However, the strong localized impacts of nutrient reductions in the York highlight the



importance of examining the spatial variability of future changes in oyster growth. It is important for managers to consider local conditions when assessing the effects of nutrient reductions on oyster production.

Increased water temperatures are projected to slow oyster growth in the future. Specifically, large reductions in tissue weight are underpinned by three primary mechanisms: limitations on filtration at high temperatures (Loosanoff, 1958), increased respiration rates (Dame, 1972), and reduced food availability. In *EcoOyster,* the optimal temperature for Eastern oyster filtration is 27°C (Cerco et al., 2005; Jordan, 1987), and under warming, the frequency at which ambient summer temperatures will surpass this optimal temperature will be higher (Fig. 7a), therefore causing more frequent declines in filtration rate (Cerco et al., 2005; Fulford et al., 2007). There is no clear optimal temperature for oyster respiration, and therefore it is assumed to increase exponentially with temperature (Hochachka and Somero, 2002). Thus, as oyster filtration rates begin to decline at high temperatures, respiration rates will continue to rise and decrease the potential for tissue accumulation (Fig. S1). Previous studies on juvenile Eastern oysters do not support a consensus on the relationship between warming and tissue growth. Some report that growth is inhibited at higher temperatures (31°C, Stevens and Gobler, 2018; 30°C, Speights et al., 2017). In contrast, Talmage and Gobler (2011) found no significant influence of high temperature (28°C) alone on tissue growth. The optimal temperature for oyster filtration may also vary among oysters, based on observations of maximum filtration rates of adult Eastern oysters occurring between 28.1°C– 32°C (Loosanoff, 1958). Variation in experimental design may have contributed to the contrast in results summarized here, in addition to the influence of local adaptation (Burford et al., 2014). Other studies that incorporate higher temperature thresholds into their models predict increases in oyster biomass under warming in Chesapeake Bay (Allen et al., 2023), underscoring the importance of properly parameterizing growth models. Due to a lack of consensus on temperature limits of Eastern oyster filtration, further research is needed to more robustly represent oyster filtration in bioenergetics models and improve predictions of impacts of warming on oysters and their ecosystem services in the region.

Warming will likely have a negative effect on food availability for oysters. Compared to the effects of nutrient reductions, warming will have a much more widespread influence on POC, causing reductions throughout the model domain (Fig. 10b,c). Despite warming increasing rates of POC production via increased phytoplankton growth rates, factors such as nutrient limitation and increased respiration rates will result in a net decrease in POC availability. In the tributaries, reductions in food availability will be most widespread due to warming, but less extreme than those from nutrient reductions in the shallow parts of the tributaries where oysters are affected. Remineralization of organic carbon in marine systems is temperature-dependent (López-Urrutia et al., 2006), and as warming occurs, remineralization of detrital carbon to DIC in bottom waters will occur at higher rates. As much of the lower bay is nutrient-limited (Zhang et al., 2021), phytoplankton growth rates will not increase much from warming alone; therefore, increased remineralization will likely reduce the overall amount of food available to oysters. Despite a similar average reduction in food availability being predicted for the future warming simulation and managed nutrient reductions simulation, the influence of warmer temperatures will amplify the negative effects of reduced food availability on growth. In this study, the critical temperature at which respiration rates exceed assimilation rates is dependent on filtration. When food availability limits filtration, this critical temperature lowers, and the temperature threshold





for tissue loss is lowered. Experimental studies have demonstrated how organic carbon may be influenced by both warming
and acidification (Simone et al., 2021), but as these dynamics can differ based on nutrient availability, it is important to consider
how climate change will influence food webs and nutrient dynamics.

The projected mid-century reductions in oyster growth obtained from this analysis are consistent with the results of
other studies that examine oyster growth under projected climate change conditions. A study modeling oyster responses in
Barataria Bay, LA, for example, predicts that under a warming and high flow scenario (though without the effects of future
nutrient reductions or atmospheric $CO_2$), oysters will experience widespread mortality in fresher parts of the bay by the end of
the century (Lavaud et al., 2021). Experimental studies have shown similar negative effects of acidification, warming, lower
food availability, and increased freshwater flow on oyster survival (La Peyre et al., 2013; Rybovich et al., 2016; Lowe et al.,
2019; Jones et al., 2019). Da (2023) found that the reductions in salinity and $\Omega_{Ca}$ that result from high discharge events in the
York River will increase in extent as climate change progresses and increasingly threaten aquaculture production. In the
Chesapeake Bay, extreme precipitation events are predicted to occur more frequently with future climate change, however an
overall decline in annual average precipitation is also predicted (St. Laurent et al., 2021). As a result, the overall impact of
freshwater from the land is not projected to change significantly in the future (Hinson et al., 2023). Changes in precipitation
were thus not simulated in this study, but future work could examine the dynamics of climate change, salinity, $\Omega_{Ca}$, and oyster
growth in a year with more heavy rainfall events but lower annual rainfall.

**4.3 Influence of future changes in oyster growth on aquaculture**

Understanding the relative impacts of global climate change and local nutrient management actions on oyster growth
and survival will allow aquaculture producers to anticipate how their oyster stock may respond to these anthropogenic changes.
As the effects of climate change are subject to natural interannual variability, the magnitude of acidification and warming in a
given year will likely differ (Cai et al., 2021; Moore-Maley et al., 2016; Li et al., 2016), influencing oyster growth through
differing mechanisms. Smaller oysters resulting from slower growing times in a particularly warm year may present a different
challenge to growers than weak-shelled oysters in a year with lower $\Omega_{Ca}$ and average temperatures. Mortality may also become
a more urgent challenge as summer temperatures warm. A previous study examining commercial performance of Pacific
oysters in Brazil found that interannual variability in temperature, chl*a* abundance, and climate events influenced survival and
growth phase timing (Mizuta et al., 2012). High temperatures inhibited survival of oyster seed in that study, which frequently
occurs in Pacific oysters (*Crassostrea gigas*) during the summer months in Europe and California (Goulletquer et al., 1998;
Burge et al., 2007; Malhan et al., 2009). A similar phenomenon has been observed in Eastern oysters; however, mortality
events in this species have not been conclusively linked to warmer water temperatures (Guevelou et al., 2019; Biranik and
Allam, 2023), and the cause is yet to be resolved for either species. Nonetheless, the increasing occurrence of spring/summer
mortality in Eastern oysters suggests that shifting the time of planting oysters on leases later in the year may help mitigate the
risk of widespread mortality, although the economic tradeoffs involved in shifting the growing season for oysters should be
taken into account.



Future climate change and nutrient management are projected to worsen conditions for oyster growth, and the spatial
variation in these changes may unevenly influence aquaculture production. While reductions in shell and tissue growth are
predicted for nearly all regions where oysters grow, these changes will likely differ based on present-day environmental
conditions. Under present day conditions, the most oyster growth is projected to occur in regions with some of the highest
present-day $\Omega_{Ca}$ and the greatest projected $\Omega_{Ca}$ reductions, i.e., in the Rappahannock River and mainstem shoals. Some of the
most dramatic tissue and shell reductions are projected to occur in the York and upper Rappahannock, where reduced food
availability and low $\Omega_{Ca}$ will limit oyster filtration and shell growth. Oysters in parts of both the Rappahannock and York
Rivers will likely face mortality (represented by near complete depletion of oyster shell and tissue) by mid-century (Figs. 8,
9). However, these reductions will not be spatially uniform, underscoring the importance of oyster farm site selection within a
tributary. In contrast, oysters grown outside the tributaries are projected to exhibit a smaller decline in growth, indicating
greater future opportunity for oyster farming in these locations. Under the business-as-usual climate change trajectory analyzed
here, bottom conditions in the tributaries will be less suitable for oyster aquaculture by mid-century, and producers might
consider alternate farm locations or shifting production methods toward floating culture to avoid exposure to low $\Omega_{Ca}$
conditions and access greater food availability.
Beyond reduced oyster growth, aquaculture operations may also be affected in the future by temporal changes in
optimal growing conditions. Due to the input of freshwater that lowers DIC and TA and increases $p$CO$_2$ (Cai et al., 2017; Cai
et al., 2021; Da et al., 2024), the greatest magnitude of $\Omega_{Ca}$ reductions occurs in spring. The majority of oyster growth is
projected to occur in the spring and summer (Fig. 7), so changes to growing conditions may be most consequential during
these warmer months. Deployment of oyster seed generally begins in the spring and continues into the summer, so it is
important for producers to be aware of ambient conditions being experienced by their newly deployed oysters. As spring
temperatures warm, phytoplankton blooms will likely occur earlier in the year, shifting the time when food availability is
highest (Da et al., 2021). Oysters deployed earlier in the year may benefit from greater food availability and perform better
than oysters deployed in July or August when waters are warmest. However, they may also face the challenge of spring/summer
mortality events, revealing the complexity of timing oyster deployment under worsening climate change conditions. For oyster
farms closer to freshwater sources, the combined effects of low $\Omega_{Ca}$, low salinity, and high summer temperatures may severely
inhibit growth and extend time-to-market.
**4.4 Future work**
Providing the aquaculture industry with the best existing estimates of climate change impacts to their operations will
allow them to make more informed decisions about their future practices. This study used a 120-meter horizontal resolution
model grid to examine near-lease-level effects of climate change and management actions on oyster growth in a section of the
lower Chesapeake Bay. Similar studies with high resolution model grids in other systems will strengthen our understanding of
how regional anthropogenic effects will influence the oyster aquaculture sector and could be used to identify areas of
opportunity for new aquaculture practices (Swam et al., 2022; Palmer et al., 2021; Lavaud et al., 2024). The present study





incorporated one Earth System Model and one emissions scenario; future work should quantify how these choices impact
estimates of future $\Omega_{Ca}$ and oyster growth (e.g., Hinson et al., 2023). Future modeling studies should also incorporate other
climate change impacts, such as sea level rise and increased storminess which are projected to influence conditions for oyster
growth in the Chesapeake Bay region (Seneviratne et al., 2012; Lowe et al., 2019, Rybovich et al., 2016, Jones et al., 2019).

To improve estimates of shell and tissue growth of oysters under climate change, additional experimental studies
should be conducted to reduce the data gaps that currently limit model formulations. Uncertainties in the functional
relationships and rate parameters used in these models may lead to an inaccurate influence of some environmental variables
on oyster growth. For example, results in this study may be particularly sensitive to the optimum temperature for filtration
rate. Reductions in tissue weight are particularly dramatic when average temperature conditions at oyster lease sites remain
above this optimal temperature from mid-June through late August, a vital time for oyster growth. As a result, growth in the
model is sensitive to the simulation start date, and future studies should compare the influence of warming on growth in
simulations that start at different times in the year. Many physiological studies of temperature impacts on oyster filtration date
back to the mid-to-late-20[th] century, and present-day seasonal extremes that coastal organisms experience may routinely exceed
the maximum temperatures used in many of these earlier experimental designs. For example, Jordan (1987) used a maximum
temperature of 27 °C, which was the ambient temperature when samples were collected in July from the Choptank River, MD.
Between 1985 and 2014, bottom waters of the north mesohaline Bay main stem (closest to the Choptank) warmed 1.01 ± 0.13
°C during May to October (Hinson et al., 2022), and the present study predicts a 1.5 ± 0.26 °C increase across the entire model
domain between 2017 and mid-century. To build stronger models of future climate impacts, and to expand scientific
understanding of physiological limits of the Eastern oyster, future studies should re-examine temperature limitations on oyster
filtration and respiration by using higher experimental temperatures.

As oyster growth is highly sensitive to food availability, improved measurements of particulate organic carbon in the
region would fortify projections of oyster production under future climate change and nutrient management. Here, it is assumed
that oysters feed on POC, a combination of plankton and detritus. However, average POC concentrations are highly
spatiotemporally variable in the Chesapeake Bay due to eutrophication and algal blooms. In this study, POC was
underestimated in the tributary channels; however, it is unclear how well POC was estimated in oyster growing areas, as *in*
*situ* measurements are currently limited to stations in the channels during monthly or semi-monthly sampling cruises. More
routine POC measurements, as well as measurements of POC in regions where oyster farming operations occur, are needed to
verify the spatiotemporal dynamics of food availability. Improved measurements of oyster food availability would allow for
stronger model skill assessment and improved projections of oyster production.
**5 Conclusions**

This study predicts widespread reductions in $\Omega_{Ca}$ in the lower Virginia tributaries of the Chesapeake Bay by mid-
century, highlighting the use of high-resolution model projections to better understand present-day carbonate chemistry
conditions and to predict the effects of climate change on a region of high interest for aquaculture production. While similar



modeling studies have projected acidification conditions in coastal regions with 3D coupled models (Siedlecki et al., 2021a,b;
Fujii et al., 2023) or modeled oyster growth with remote-sensing data and dynamic energy budget models (Palmer et al., 2020;
Palmer et al., 2021; Bertollini et al., 2021), the present study projects both carbonate chemistry conditions and oyster
bioenergetics in the Chesapeake Bay with the highest resolution thus far. Specifically, widespread reductions in $\Omega_{Ca}$ will
negatively impact oyster growth, with implications for aquaculture operations and local and regional economies. As bottom
conditions worsen, altered site-selection for oyster farms or other adaptive measures will become imperative to sustain
production and reduce the impacts of low $\Omega_{Ca}$ on farmed oysters.
Increased atmospheric $CO_2$ and nutrient reductions are projected to inhibit oyster calcification, while warming and
nutrient reductions will reduce oyster tissue and shell growth due to limitations on filtration and lowered food availability.
While the effects of global climate change on oyster growth are projected to be much stronger overall than the effects of local
nutrient management, lowered food availability from nutrient reductions may have a strong influence on oyster growth in
certain parts of the study region. As a result, all areas will not be equally vulnerable to future changes in the atmosphere and
watershed. Understanding how individual drivers influence oyster growth is important for predicting effects on aquaculture
production in the context of interannual variability of climate change and nutrient management outcomes. While the negative
effects of temperature on growth were strong in this study, additional studies on Eastern oyster temperature limits are needed
to improve projections, particularly as summer mortality of oysters is already common. Increased *in situ* measurements of
biogeochemical variables and experimental studies on oyster physiology and bioenergetics will allow for improved projections
of mid-century conditions and their potential impacts on oyster growth and the aquaculture industry.

**Code Availability**
Model code will be available upon request.

**Data Availability**
Model output will be available with a DOI on William & Mary ScholarWorks.

**Author Contribution**
MAMF, EBR, MJB, and PS wrote the proposal and acquired the funding for the project; MAMF, PS, and FD developed the
ROMS-ECB code; MJB developed the EcoOyster code; CC, MAMF, and EBR designed the experiment; CC ran model
simulations, analyzed the output, created the figures, and wrote the manuscript draft; MAMF, EBR, MJB, PS, and FD reviewed
and edited the manuscript.

**Competing Interests**
The authors declare that they have no conflict of interest.



**Acknowledgements**
This paper is the result of research funded by the National Oceanic and Atmospheric Administration's Ocean Acidification
Program under award NA18OAR0170430 to the Virginia Institute of Marine Science (VIMS). We would like to thank the
VIMS and William & Mary (W&M) high performance computing group for their technical support and computing resources.
We also thank the NOAA Chesapeake Bay Program for providing us with watershed inputs from their Phase 6 Watershed
Model. This work was largely made possible by Sara Blachman's hard work on the EcoOyster model. We thank Bill Walton
for his aquaculture expertise and support throughout the project. The model results used for this manuscript are available at
W&M Scholarworks.

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
