# Peer review of "Acidification, warming, and nutrient management are projected to"

_EGUsphere, 2024_

## Referee Comment (RC2)

This study investigates the effects of future acidification, warming, and nutrient management on Eastern oysters in the lower Chesapeake Bay. The authors developed an oyster bioenergetics model integrated into a 3-D hydrodynamic-biogeochemistry model to simulate and compare present-day conditions with projected mid-21st century scenarios, focusing on increased atmospheric $CO_2$ levels, temperature rise, and managed nutrient reductions. The model results identify areas where oysters are most vulnerable to these future conditions and quantify the combined and respective impacts of the three stressors on oyster growth. These findings offer helpful insights for stakeholders seeking to understand and mitigate these impacts.

Overall, I think the manuscript makes a valuable contribution to the field. The methodology is sound, and the interpretation of results is generally well done. However, there are several areas for improvement, mainly concerning model description, validation, and the presentation of some results. My detailed comments are as follows.

**1. Title**

The study identifies warming as one significant stressor that negatively impacts oyster tissue and shell growth. However, the title currently highlights only acidification and nutrient management. Should it also incorporate 'warming' to more accurately reflect the findings?

**2. Introduction**

The information provided in the Introduction is comprehensive; however, the first half could benefit from a bit more cohesion to improve the flow and enhance engagement. For example, it first introduces the global problem of oceans absorbing too much $CO_2$, then narrows down to emphasize that estuaries are suffering more from acidification; next, it shifts to the global issue of oceans absorbing excess heat and warming, and finally returns to coastal waters again to discuss the impact of warming on coastal acidification.

**3. Model description**

My major comment on model description is that it lacks some critical information that would help readers understand the method and replicate the experiments. Below I list specific areas that require clarification or elaboration.

L130-133: The model is N-based, with phytoplankton and zooplankton C computed using the Redfield ratio. What ratio is used for semi-labile and refractory dissolved organic nitrogen (DON)? Is it also based on the Redfield ratio? Should the carbon-to-nitrogen ratio for refractory DON differ from that of other organic nitrogen? Additionally, why are both small and large detritus simulated in terms of nitrogen and carbon instead of assuming a fixed stoichiometry?

L135-136: Nitrification/denitrification and remineralization also occur in other vertical layers. They should not be categorized as "additional" biogeochemical sources and sinks exclusive to the bottom vertical layer.

L138: How is TSS calculated within the model? It's not listed as a variable in the ECB model according to L130-131.

L139-140: Does the sediment transport module account for the sediment-water exchange processes?

L167-168: The analysis focuses on the deepest vertical level of the model. Are oysters cultivated at the bottom of this region?

L171-172: Similarly, are diploid oysters the primary aquaculture form in this region?

L173: What constitutes POC? The ROMS-ECB model includes phytoplankton, zooplankton, and small and large detritus. Are all these components considered in the calculation of POC, which serves as food for oysters?

L162-193: "2.1.3 Oyster bioenergetics model" – I find it challenging to follow the description. The EcoOyster model is referred to as a bioenergetics model; however, it's unclear how the model is formulated in terms of energy flow and how it is linked to the biogeochemical model in terms of mass. While the supplement includes a list of equations, it doesn't provide explanations. For example, in Tables S1 and S2, what does the subscript j refer to? In Equations 7 and 8, how are $w_j^{gonad}$ and $K_{gonad}$ defined?

L284-287: Please clarify how the watershed inputs are specifically adjusted to meet the "nutrient reduction goals."

L306-307: Are the locations identified by model outputs also where most existing oyster farms are situated?

**4. Model validation**

How well does the EcoOyster model simulate the observed temporal variations in oyster growth parameters, such as shell weight and tissue weight?

**5. Presentation**

Except for the first paragraph, Sections 2.3 should be moved to the Results.

Figure 2 – the caption duplicates that of Figure 1.

Figure 5 – the x-axis is missing; the figure caption is incomplete, lacking references to panels e and f for oxygen and TSS.

**6. Other minor comments**

L16: The specific future scenario employed in this study should be mentioned, as the chosen scenario will significantly influence the impact assessment.

L37-38: Suggest briefly explain what "ocean water undersaturated with calcite" means for calcifying organisms.

L151: "minimum" phytoplankton growth rate – should it be corrected to "maximum?"

LL489-492: This study uses the worst climate scenario. Does this contribute to the estimated faster acidification rate?

L501: "Eutrophication can suppress acidification by increasing primary production" – This is accurate for surface waters, but it does not hold true for bottom water.

L536-662: Given the critical role of optimal temperature, have sensitivity experiments regarding this temperature threshold been conducted?

L569-573: In the cited study (Lavaud et al., 2021), does warming affect oyster growth through the same mechanisms as in this study? Providing this information would clarify whether the findings of the two studies are "consistent."

L622: "mortality events" because of what?

L641-643: Got confused here. Why does the model simulation start date matter? Are you referring to the starting date of oyster cultivation?

---

## Author Comment (AC1)

Reviewer #2 Response

*We thank the reviewer for their careful reading of the manuscript and for providing insightful comments that will improve the paper for future readers. Reviewer comments are in black while the author responses are in blue.*

This study investigates the effects of future acidification, warming, and nutrient management on Eastern oysters in the lower Chesapeake Bay. The authors developed an oyster bioenergetics model integrated into a 3-D hydrodynamic-biogeochemistry model to simulate and compare present-day conditions with projected mid-21st century scenarios, focusing on increased atmospheric $CO_2$ levels, temperature rise, and managed nutrient reductions. The model results identify areas where oysters are most vulnerable to these future conditions and quantify the combined and respective impacts of the three stressors on oyster growth. These findings offer helpful insights for stakeholders seeking to understand and mitigate these impacts.
Overall, I think the manuscript makes a valuable contribution to the field. The methodology is sound, and the interpretation of results is generally well done.

*We thank the reviewer for these positive comments.*

However, there are several areas for improvement, mainly concerning model description, validation, and the presentation of some results. My detailed comments are as follows.

**1. Title**
The study identifies warming as one significant stressor that negatively impacts oyster tissue and shell growth. However, the title currently highlights only acidification and nutrient management. Should it also incorporate 'warming' to more accurately reflect the findings?

*This is an excellent point. Warming was a more important stressor than we had originally thought, and we agree that this should be added to the title. The title will be modified to: "Acidification, warming, and nutrient management are projected to cause reductions in shell and tissue weights of oysters in a coastal plain estuary"*

**2. Introduction**
The information provided in the Introduction is comprehensive; however, the first half could benefit from a bit more cohesion to improve the flow and enhance engagement. For example, it first introduces the global problem of oceans absorbing too much $CO_2$, then narrows down to emphasize that estuaries are suffering more from acidification; next, it shifts to the global issue of oceans absorbing excess heat and warming, and finally returns to coastal waters again to discuss the impact of warming on coastal acidification.

*Thank you for this suggestion. The introduction will be restructured so that the global phenomena are introduced before narrowing it down to coastal systems.*

**3. Model description**
My major comment on model description is that it lacks some critical information that would help readers understand the method and replicate the experiments. Below I list specific areas that require clarification or elaboration.

*Thank you for this suggestion. We wanted to keep the paper as succinct as possible, but it is important that readers understand the methodology, so we will add more details to the methods as suggested.*

L130-133: The model is N-based, with phytoplankton and zooplankton C computed using the Redfield ratio. What ratio is used for semi-labile and refractory dissolved organic nitrogen (DON)? Is it also based on the Redfield ratio? ***Should the carbon-to-nitrogen ratio for refractory DON differ from that of other organic nitrogen?***

While the C:N ratio for particulate organic matter is fixed at the Redfield ratio, the ratio for dissolved organic matter within the estuary is allowed to freely evolve in time since, as the reviewer correctly points out, DOC:DON is typically not in the Redfield ratio (refractory organic matter has a considerably higher C:N ratio). We will make this clearer in the revised manuscript.

Additionally, why are both small and large detritus simulated in terms of nitrogen and carbon instead of assuming a fixed stoichiometry?

Although we are indeed using a fixed stoichiometry for particulate organic C and N now, having separate compartments for detrital C and N will facilitate the implementation of a varying stoichiometry version of the model in the future.

L135-136: Nitrification/denitrification and remineralization also occur in other vertical layers. They should not be categorized as "additional" biogeochemical sources and sinks exclusive to the bottom vertical layer.

Excellent point. The sentence will be edited as suggested.

L138: How is TSS calculated within the model? It's not listed as a variable in the ECB model according to L130-131.

TSS is calculated within the model as the sum of the four inorganic sediment size classes and particulate organic matter (which is defined as the sum of small detritus, large detritus, phytoplankton and zooplankton). This will be added into the methods text in Section 2.1.2.

L139-140: Does the sediment transport module account for the sediment-water exchange processes?

Yes, the sediment transport module accounts for sediment-water exchange processes, i.e., it includes both deposition and resuspension of inorganic sediment and particulate organic matter. This will be clearly stated on lines 139-140 to avoid any confusion.

L167-168: The analysis focuses on the deepest vertical level of the model. Are oysters cultivated at the bottom of this region?

Yes, oysters are frequently cultivated at the bottom in these tributaries. Following line 168, we will add: "This is representative of conditions similar to on-bottom or bottom cage aquaculture methods that are common in Virginia."

L171-172: Similarly, are diploid oysters the primary aquaculture form in this region?

This is an excellent point. Diploid oysters are less commonly used in shellfish aquaculture in this region than triploid oysters; however, we simulated the responses of diploid oysters because the laboratory experiments used to develop the EcoOyster model parameterizations were conducted on diploids.

This is addressed in the current manuscript methods text (171-172): "For the purpose of this study, only diploid oysters were included, as the triploid allometric equations are not as well constrained." We will revise this text for additional context: "For the purpose of this study, only diploid oysters were included, **as model equations were developed from a study on diploid oysters** and triploid allometric equations are not as well constrained."

We will also add the following text: "This study focused on diploid oysters, since the laboratory experiments used to develop the EcoOyster model parameterizations were conducted on diploids. Future laboratory experiments with triploids would enhance our modeling efforts for triploid oysters, which are commonly used in aquaculture in the Chesapeake Bay region."

L173: What constitutes POC? The ROMS-ECB model includes phytoplankton, zooplankton, and small and large detritus. Are all these components considered in the calculation of POC, which serves as food for oysters?

Yes, we absolutely should have made this clearer in the text. We will add a sentence in the methods stating: "Particulate organic carbon (POC) is calculated as the sum of phytoplankton carbon, zooplankton carbon, and small and large detrital carbon."

L162-193: "2.1.3 Oyster bioenergetics model" – I find it challenging to follow the description. The EcoOyster model is referred to as a bioenergetics model; however, it's unclear how the model is formulated in terms of energy flow and how it is linked to the biogeochemical model in terms of mass.

The bioenergetics model (EcoOyster) includes equations for shell and tissue weight as a function of a number of environmental variables. EcoOyster is one-way coupled with ROMS-ECB (see line 164) such that environmental and water quality variables from ROMS-ECB are used to force EcoOyster at every time step. The physical, biogeochemical and oyster equations are all solved at the same time (as opposed to running the three models separately and after the fact, using one to force another.) However, because the coupling is only one-way, the simulated oysters do not have an effect on the environmental variables, i.e. they do not improve water clarity or impact carbonate cycling. We will make this one-way linkage between EcoOyster and ROMS-ECB clearer in the revised manuscript.

While the supplement includes a list of equations, it doesn't provide explanations. For example, in Tables S1 and S2, what does the subscript j refer to? In Equations 7 and 8, how are wjgonad and Kgonad defined?

Thank you for pointing out these issues. The subscripts were an oversight, remaining from a previous version of the model equations (appearing in the thesis); these will be removed. $W_{gonad}$ is defined in equation 2. $K_{gonad}$ is the Michaelis constant for the resorption function (equation 7), which will be clarified in a footnote for table S2.

L284-287: Please clarify how the watershed inputs are specifically adjusted to meet the "nutrient reduction goals."

Gopal Bhatt, who runs the CBP Phase 6 Watershed Model, provided us with the watershed model output that they use in their scenarios when assuming the nutrient reductions goals (TMDLs) have been met. Meeting the TMDLs mandates a total nitrogen load reduction of 25% compared to 2009 levels. Because in this study our baseline is 2017 rather than 2009, our

percent load reductions are slightly different between our reference and TMDL runs, however the actual loads in our TMDL runs are very similar to those used in the EPA's regulatory TMDL scenarios.

The 25% reduction of total nitrogen will be added as follows, to make this clearer: "Future watershed inputs for the *TMDL* and *Combined Future* simulations included a climatology of nitrate, ammonium, dissolved organic matter, and particulate organic matter concentrations, derived from a Phase 6 CBPWM 1991-2000 run using reduced nutrient concentrations assuming the TMDLs had been successfully achieved (**essentially a 25% reduction in total nitrogen loading;** Bhatt et al., 2023)."

L306-307: Are the locations identified by model outputs also where most existing oyster farms are situated?

This is a very good question. It turns out that the majority of the coastline is leased for aquaculture, but not all leases are in locations where environmental conditions are conducive for oyster growth. Thus, just because an area is leased, does not mean oyster aquaculture is occurring there (some people buy leases to prevent others from establishing aquaculture farms in the waters offshore of their property); unfortunately, we are not authorized to report the location of aquaculture activity due to privacy concerns. However, we do know that the grid cells that report oyster growth are generally consistent with the location of active oyster farms, so we will add this text as a last sentence in the methods: "results are shown for grid cells characterized by environmental conditions that support oyster growth, which are inclusive of locations of active oyster farms. (Specific areas of active aquaculture activity cannot be reported here due to privacy concerns.)"

**4. Model validation**
How well does the EcoOyster model simulate the observed temporal variations in oyster growth parameters, such as shell weight and tissue weight?

This is addressed at the end of section 2.3, after we describe how growth rates determined from the EcoOyster equations were compared with in situ data: The resulting shell growth predicted by the model was found to be close to the in situ data (52.0 ±1.1 mm y$^{-1}$ and 51.3 ± 0 2.9 mm y$^{-1}$ for the model and observation means and standard deviations, respectively).

**5. Presentation**
Except for the first paragraph, Sections 2.3 should be moved to the Results.

Thank you for this comment! Reviewer 1 agreed. This will be moved to the Results section.

Figure 2 – the caption duplicates that of Figure 1.

Thank you for pointing this out! The duplicate text will be deleted.

Figure 5 – the x-axis is missing; the figure caption is incomplete, lacking references to panels e and f for oxygen and TSS.

We apologize for this error. Unfortunately, the figures were accidentally cropped just before uploading, when we moved the figures from the end of the document to the location in the document where the figures were first mentioned. This will be fixed, and the caption corrected to include the last two panels, i.e., "(e) oxygen, and (f) TSS."

**6. Other minor comments**

L16: The specific future scenario employed in this study should be mentioned, as the chosen scenario will significantly influence the impact assessment.

We agree and apologize for the omission. The text here in bold will be added: "Model simulations were forced with projected future conditions (assuming mid-21st century atmospheric **CO₂** and temperature **under emissions scenario RCP 8.5,** as well as managed nutrient reductions) and compared with a realistic present-day reference run."

L37-38: Suggest briefly explain what "ocean water undersaturated with calcite" means for calcifying organisms.

A sentence will be added here, specifying, "which elevates the energetic costs of shell-building for calcifying organisms."

L151: "minimum" phytoplankton growth rate – should it be corrected to "maximum?"

Thank you for catching this error; this indeed should be "maximum". This will be updated in the text.

LL489-492: This study uses the worst climate scenario. Does this contribute to the estimated faster acidification rate?

This is a great question! Although using RCP8.5 will result in a higher acidification rate compared to using RCP4.5, in this case the text is comparing our results to Siedlecki et al. (2021a) who also used RCP8.5. They also used a similar methodology (the delta method) to apply these changes to reference forcings.

We will update the text as denoted in bold here: "Comparing our results to other studies examining the effects of acidification reveals that the Chesapeake Bay will likely acidify faster than the US West Coast. Siedlecki et al. (2021a) projected a decrease of 0.8-1.0 in $\Omega_{Ca}$ in the Northern California Current System between 2000 and 2100 **under the same future emissions scenario used in our study, i.e., RCP8.5"**

L501: "Eutrophication can suppress acidification by increasing primary production" – This is accurate for surface waters, but it does not hold true for bottom water.

This will be addressed by adding "in surface waters."

L536-662: Given the critical role of optimal temperature, have sensitivity experiments regarding this temperature threshold been conducted?

Not yet. The authors identify this as a major limitation of the study, and discuss this at length in section 4.2: Future projections for oyster growth. We suggest additional studies on temperature limits in this section and in and section 4.4: Future work, and mention this in section 5: Conclusions.

L569-573: In the cited study (Lavaud et al., 2021), does warming affect oyster growth through the same mechanisms as in this study? Providing this information would clarify whether the findings of the two studies are "consistent."

Lavaud et al. (2021) did not test the effects of temperature alone, but conducted simulations where temperature is a co-stressor with low salinity. They also didn't measure the exact same metrics: they have shell length, total wet weight, and reproduction, and found that while low salinity worsened all of those, a coincident temperature increase of 2.6 °C led to 100% mortality. Higher mortality rates were predicted in lower salinity areas.

The 'consistency' we mention in this sentence is referring to the combination of warming and low salinity being threatening for oysters (i.e., lower growth in fresher parts of the bay). We will clarify in the text that 'consistency' is specifically referring to the combination of warming and low salinity.

L622: "mortality events" because of what?

This is a good question! As of right now, the cause of these mortality events has not been conclusively determined, but summer mortality events are a significant issue in the oyster aquaculture industry and currently of high research/management interest. We will edit the manuscript to make it clearer that this is an area of active research.

L641-643: Got confused here. Why does the model simulation start date matter? Are you referring to the starting date of oyster cultivation

It is possible that the model simulation may be sensitive to the time when the simulation (growth) begins, since if oysters are deployed later in the summer they will not have as much time to grow before temperatures drop below the optimal threshold. This is likely not a major effect and not one we had time to fully address. Because this paragraph is more about additional experimental studies that should be conducted, we have chosen to remove this confusing sentence.

---

## Author Response (AR2)

**Response to editor's comments on Czajka et al., 2025, Version 2:**

We thank the editor and reviewer for their second reading of the manuscript, and their additional comments that will further improve the manuscript. Once again, reviewer comments are in black, and our author responses are in blue.

Thank you for your thorough responses and revision. The manuscript has been re-evaluated by one of our previous reviewers, who was very positive about the revision and requested clarification on two minor points related to the model settings. I have also reviewed the revision and provide the following technical comments for your consideration:

Thank you for your additional efforts on this manuscript!

L46: Suggest replacing "such as the input of acidic freshwater and nutrient runoff from precipitation" with "such as the inflow of acidic freshwater and the runoff of nutrients from precipitation" to better distinguish these as separate processes.

Done.

L176: "as model equations were developed from a study on diploid oysters " - please cite the study from which these model equations were derived.

Unfortunately, this study is not quite yet out in print but we reference our Ocean Sciences abstract: Rivest et al., 2020.

Table 3. May consider replacing identical values in simulations that match either the reference run or combined future run with the labels 'Reference' or 'Future.' This would reduce redundancy and help readers quickly identify which variables differ across simulations. For example, since simulation AtmCO2 mirrors the reference run in all variables except $\Omega_{Ca}$, only $\Omega_{Ca}$ would display a numerical value, while other cells could simply state 'Reference.'

This is a fabulous idea in principle, but unfortunately is difficult to implement. The idea is clear for the example given (AtmCO2) but in most cases for the other simulations, we cannot simply use "future" or "reference". This is because we are referring to model results in this table, not model forcings. For example, the TMDL simulation certainly alters oxygen, but the effect is so small, it looks like the oxygen is the same as in the "reference" case. Although we like the idea in principle, we feel the best option is to retain the original version of Table 3.

Figure 5. Suggest increasing the thickness of the dashed line for better visibility.

Done.

In the caption, correct "$\Omega Ca$" to $\Omega_{Ca}$ (subscript formatting).

Done.

Figure 8. Suggest capping the y-axis maximum at 0.

Done.

Figure 9 caption: "change in bottom POC"

Done.

Figures 10 & 11 caption: "(a,d) AtmCO2, (b,e) Temp, and (c,f) TMDL"

Done.

Upon addressing these minor points, we will be pleased to accept your manuscript for publication. Thank you for choosing Biogeosciences to share your valuable work.

Thank you!

**Response to reviewer's comments on Czajka et al., 2025, Version 2:**

The authors have thoughtfully addressed my previous comments, and I recommend acceptance after clarifying the following minor points:

Thank you for your additional time reviewing the manuscript.

1. In response to my earlier question about the C:N ratios for semi-labile and refractory DON, the authors state that these ratios "are allowed to freely evolve with time" (Lines 130–131). While I agree that these ratios should vary, I'm still confused about how this temporal variability is implemented in the model.

Thank you for this comment! We now realize that the confusion lies in the fact that we didn't mention that we had DOC state variables as well as DON state variables. Because they are separate state variables, they evolve separately in time and their ratio varies. We have fixed this in the text by adding the bolded text below:

"small and large detrital nitrogen and carbon, semi-labile and refractory dissolved organic nitrogen **and carbon**, DIC, TA, and dissolved oxygen ($O_2$)."

2. The authors' explanation of the one-way coupling between EcoOyster and ROMS-ECB is helpful. However, I'm still confused about the model being described as a "bioenergetics model" when its governing equations (provided in the Supplement) appear to be formulated in terms of mass balances rather than energy flows. Could you help me understand where the energy flow calculations come into play? Is that handled in a different part of the model that I'm missing?

This is a very good point. Because this is technically not a bioenergetics model, we have removed "bioenergetics" from the text and simply call this an "oyster growth model".